# Mechanism of Topology Change of Flat Magnetic Structures

**DOI:** 10.3390/e24081104

**Published:** 2022-08-11

**Authors:** Eugene Magadeev, Robert Vakhitov, Ildus Sharafullin

**Affiliations:** Laboratory of Design of New Materials, Bashkir State University, 450076 Bashkortostan, Russia or

**Keywords:** perforated film, ferromagnet, anisotropy of the “easy-plane” type, antidot, bound states, current impulse, quasiparticles, topological charge

## Abstract

The paper investigates the processes of the magnetization reversal of perforated ferromagnetic films with strong anisotropy of the easy-plane type. The investigations have shown that, influenced by a current impulse passing through an antidot, an inhomogeneous magnetic structure is formed, which is accompanied by the localization of a quasiparticle with the +1 topological charge on the antidot and by an emission of a quasiparticle with a –1 charge. It is established that this scenario of the film magnetization reversal underlies a reformation of its inhomogeneous structure also if two or four antidots are present in the film, irrespective of the fact of through which antidots and in which directions the currents are passed. The results of the research obtained by using two independent methods (solving the Landau–Lifshitz–Gilbert equations and analyzing the lattice model) demonstrated good agreement between the two. It is shown that a magnetic film comprising two or four antidots can be used as a memory cell for recording data in the ternary system.

## 1. Introduction

Vigorous research is currently underway to develop new-generation data recording devices which would meet the latest requirements of the units of such a kind. Evidently, they have to feature random access, non-volatility, great data density, high speed, etc. [1,2]. The most promising in this respect are the devices based on magnetic films, in which vortex-like magnetic inhomogeneities can originate (for example, magnetic vortices, skyrmions, bimerons, and magnetic bubbles [3,4,5,6,7,8]). There are certain difficulties, however, which stand in the way of creating such devices. These challenges are of various kinds, and they are yet to be fully overcome [7,9]. For instance, the development of memory devices based on magnetic bubbles [8], which have been carried out for more than a quarter of a century, lost their relevance in the early 1990s since they could not withstand competition with semiconductor technologies (due to high cost of bubble materials).

Magnetic vortices, first discovered in [9], have been studied in sufficient detail in the last two decades [10], but to date, interest in them has decreased. At the same time, there are certain signals about their possible use in spintronic devices [11,12], but not on a large scale, as previously assumed.

Magnetic skyrmions, first discovered in non-centrosymmetric magnetic materials (chiral magnets [13]), are stabilized due to the presence of the Dzyaloshinskii–Moriya interaction in them [14]. This interaction, which is very weak under normal conditions, becomes significant only at low temperatures. In addition, magnetic skyrmions can exist as stable structures in ultrathin magnetic films epitaxially grown on heavy metal substrates [4,5], as well as in Co/Pt multilayer films [15]. Further studies have shown that in these cases, the Dzyaloshinskii–Moriya interfacial interaction, which takes place in these magnets, stabilizes the magnetic skyrmions, even at room temperatures. However, certain difficulties arise in maintaining a fine balance of the interactions affecting the stability of skyrmions at such small film thicknesses (~1 nm). Accordingly, alternative ways of stabilizing skyrmions in non-chiral magnets have also been proposed [7,16]. In particular, it was shown that the formation of magnetic skyrmions is possible in uniaxial ferromagnetic films with spatially modulated material parameters. Moreover, it was found that if a uniaxial multilayer film of the Co/Pt type is subjected to local irradiation of its surface with a beam of He^+^ ions, then magnetic skyrmions can be observed in the irradiated areas (due to a uniaxial anisotropy constant decrease in them) [17,18], which are stable in wide ranges of temperatures and magnetic fields [17,19]. However, the possibility of the practical use of such materials remains open. It should be noted that the considered types of skyrmions are not limited to the above list since there is a whole family of them (“zoo” of skyrmions [5]): skyrmion, antiskyrmion, biskyrmion, skyrmionium, chiral bobber, hedgehog, etc., as well as *k*π–skyrmions (*k* = 1, 2, 3, …; π-skyrmion is an ordinary skyrmion) [20]. They differ in polarity, chirality, topological charge, type of magnetization distribution (Bloch or Neel) and the number of half-turns *k* of the magnetization vector when the radial variable changes from 0 to ∞ (if the turn is 180°, then *k* = 1, etc.). At the same time, a new approach to this problem was treated in paper [21], which proposes to employ perforated magnetic films in which flat vortex-like micromagnetic structures may form, such structures being localized in the area of two or more nearby antidots. Such inhomogeneities, being basically metastable in the absence of external influences, become stable when exposed to currents, which opens a potential for using perforated magnetic films as re-recordable super-density data storage media. The paper investigates the mechanism of magnetic structure transitions between the states with a differing topology based on which the conditions are identified at which the processes of re-recording can be implemented on a practical level.

## 2. Motion Equations

Let us consider a sample in the form of a thin ferromagnet film featuring strong uniaxial anisotropy of the “easy plane” type. Magnetic materials with such anisotropy are not uncommon; in particular, they include binary compounds of some metals with Co (e.g., the constant of uniaxial anisotropy for NdCo_5_ is equal to –4 × 10^7^ J/m^3^ [22]). Its presence results in that the spins practically do not leave the film plane [21]. We shall present the unit vector of magnetization and the effective magnetic field as $m=m_{p}+m_{n}$ and $H=H_{p}+H_{n}$, respectively, where the “*p*” index denotes the components which lie within the film plane, and the “*n*” index denotes the components that are normal in the plane. Then $\left|m_{p}\right|\approx 1$, $\left|m_{n}\right|\ll 1$, and the Landau–Lifshitz–Gilbert equation [23].
(1)$$\frac{\partial m}{\partial t}=\gamma \left[H\times m\right]+\alpha \left[m\times \frac{\partial m}{\partial t}\right]$$
where *γ* is gyromagnetic ratio, and *α* is the dissipation parameter, breaks down into the following two equations, all insignificant terms dropped in the first one:(2a)$$\frac{\partial m_{p}}{\partial t}=\gamma \left[H_{n}\times m_{p}\right]$$
(2b)$$\frac{\partial m_{n}}{\partial t}=\gamma \left[H_{p}\times m_{p}\right]+\alpha \left[m_{p}\times \frac{\partial m_{p}}{\partial t}\right]$$

Neglecting $\frac{\partial m_{n}}{\partial t}$ in the second equation and inserting $\frac{\partial m_{p}}{\partial t}$ from the first one, we have
(3)$$H_{n}=-\frac{1}{\alpha }\left[H_{p}\times m_{p}\right]$$
from where
(4)$$\frac{\partial m_{p}}{\partial t}=-\frac{\gamma }{\alpha }\left[\left[H_{p}\times m_{p}\right]\times m_{p}\right]$$

The ratio thus arrived at will acquire a simpler form if one introduces the angle *θ*, which sets the orientation of the vector $m_{p}=\left(\cos\theta ,\sin\theta ,0\right)$ in this point of the plane and also presents $H_{p}$ through the variational derivative $-\frac{1}{M_{s}}\frac{\delta E}{\delta m_{p}}$, where the functional *E* corresponds to the full energy of the magnet [23] (for clarity, however, *E* will not include the terms which are responsible for uniaxial anisotropy, which in no way impacts the reasoning):(5)$$\frac{\partial \theta }{\partial t}=-\frac{\gamma }{\alpha M_{s}}\frac{\delta E}{\delta \theta }$$

As seen from the equation of motion obtained, the dynamics of the flat magnetic structures is determined primarily by the dissipative processes. It can be demonstrated that the applicability condition of this approximation is set by the ratio $\alpha \gg \sqrt{M_{s}\left|H_{p}\right|/\left|K\right|}$, where *K* in the constant of uniaxial anisotropy. This ratio is automatically satisfied with |*K*| → ∞ for any values of α. However, with intermediate values of *K*, a term may appear in (5) which contains the second time derivative of *θ* and which gives rise to inertia, which, however, we will agree to disregard.

In the case of discrete models [24], the magnet energy *E* is a function of the limited number of angles *θ_i_*, which determine orientations of certain magnetic moments. If these spins are distributed over the lattice in a uniform fashion (disregarding maybe the areas of the sample which are occupied by the antidot), then, in a manner similar to (5), we can write the following motion Equations:(6)$$\frac{d\theta_{i}}{dt}=-\beta \frac{\partial E}{\partial \theta_{i}}$$

The coefficient $\beta \;\sim \;\alpha^{-1}$ which is contained here assigns the time scale of the processes and can be made equal to the value of 1 by selecting the appropriate units of measurement.

## 3. Base Mechanism of Topology Change

Let us consider an infinite and uniformly magnetized film with a perforation which has a straight conducting filament passing through it, perpendicular to the aforementioned plane. If a current is passed along the conductor, magnetization at each point of the film will reorient along the magnetic field lines. It can be easily noted that this process is connected with a change in the magnetic structure topology. Indeed, the angle *θ*, when circling around the hole in the initial state (*θ* = 0), remains unchanged, whereas in the final state (*θ* = *φ*, if the polar coordinate system *r*, *φ* is selected in such a way that the coordinate origin point coincides with the antidot center) changes by 2π. Introducing the notion of the topological charge of the area [5,21] as a number of turns made by the magnetization vector as it circles this area around along the boundary in a fixed direction, we can state that in the process of magnetization reversal the antidot’s topological charge changes from 0 to +1. It is evident that in this situation, the angle *θ* may remain a continuous function of spatial coordinates at an arbitrary point of time. To be able to trace the dynamics of the emerging discontinuities, we will revisit the motion equations earlier obtained.

In those cases when the effect of the demagnetizing fields in the films can be neglected, its energy, as a rule, can be presented in the following manner [23]:(7)$$E=\int \left[A{\left(\nabla \theta \right)}^{2}+\Phi_{e}\left(\theta ,r\right)\right]dV$$
where integration is performed over the whole volume of the samples, and *A* is the exchange parameters, whereas *Φ_e_* is a certain function of the angle *θ* and the spatial coordinates which characterize the external influence. Then, from (5), we obtain
(8)$$\frac{\partial \theta }{\partial t}=\frac{\gamma }{\alpha M_{s}}\left(2A∆\theta -\frac{\partial \Phi_{e}}{\partial \theta }\right)$$

It is easy to see that in the absence of external influences, Equation (8) is that of heat transfer, thanks to which a relaxation is ensured of the magnetic structures in the conditions meeting the Laplace equation ∆*θ* = 0, these conditions having been investigated in detail in [21]. All topological peculiarities of such stationary conditions strictly reflect the topology of the sample itself: in particular, it is only the areas which correspond to the holes that can possess a non-zero topological charge. An external influence is therefore the factor which brings about a change of the magnetic structure topology, whereas the exchange interaction plays but a stabilizing role by smoothing and eventually eliminating discontinuities of the function of *θ*, which arise as a side effect of the magnetization reversal processes.

In view of the above, let us assume in (8) that *A* → 0 and $\Phi_{e}=-\frac{IM_{s}}{2\pi r}\cos\left(\theta -\varphi \right)$ which corresponds to an interaction of the magnet with the field which is produced by the current *I* (here and below, arbitrary units are used for the magnitude of the electric current; recall also that the angle *φ* is one of the coordinates of the polar system). Then
(9)$$\frac{\partial \theta }{\partial t}=-\frac{\gamma I}{2\pi \alpha r}\sin\left(\theta -\varphi \right)$$

Solving this equation with the initial condition *θ* = 0, we obtain
(10)$$\theta =\varphi -2\arctan\left[\tan\left(\frac{\varphi }{2}\right)\exp\left(-\frac{\gamma It}{2\pi \alpha r}\right)\right]$$

Ratio (10) describes the dynamics of magnetization reversal of the entire film with the exception of the points lying on the half-line *φ* = π. Let us investigate the behavior of the angle *θ* in the vicinity of this half-line, specifically, on the two closely located parallel lines *r* sin *φ* = ±*h*, *h* = const. Figure 1 shows the dependence of *θ* on the coordinate *x* = –*r* cos *φ* on the lines at successive moments of time *t*, and, notably, the time lapse between subsequent blue curves *θ*(*x*) is 10 times greater than that between subsequent red ones, which in its turn is 10 times greater than the time lapse between subsequent violet curves *θ*(*x*). It can be seen from the figure that the magnetization reversal process breaks down into two stages. Initially (violet and red curves), a magnetization reversal nucleus forms near the conductor rather quickly within which *θ* varies from 0 to ±π and back. Going forward, however, the right-side part of the nucleus starts moving away from the conductor (blue curves), roughly preserving its shape. At any moment of time, this right-side part happens to be localized in a small part of the space, to which the –1 topological charge is related (see, for instance, the area in Figure 1, which is bounded by the green dashed lines), and consequently this right-side part can be identified with some quasiparticle. The left-side part of the nucleus, in a similar manner, can be identified with a fixed quasiparticle which carries the +1 topological charge. In these terms, the first stage of the magnetization reversal represents an origination of a pair of quasiparticles in the antidot area, whereas the second one is an emission of one of them, which, while gradually moving off to infinity, takes the −1 topological charge away with it, thanks to which there happens a change of the sample’s topology in general.

The mechanism described becomes even more clear if we consider a discrete model comprising 400 spins which takes into account the exchange interaction among other things. We will select the system energy as [21]
(11)$$E=\underset{i,j}{\sum }\left(1-\cos\left(\theta_{i}-\theta_{j}\right)\right)-\underset{i}{\sum }\frac{I}{r_{i}}\cos\left(\theta_{i}-\varphi_{i}\right)$$
where the first summation is performed over all of the pairs of lattice points which are the nearest neighbors, horizontally or vertically, and *r_i_*, *φ_i_* are the coordinates of the lattice point related to the spin, with the orientation determined by angle *θ_i_*. Therefore, the first term in Formula (11) corresponds to the exchange interaction (generally speaking, it should also include a factor characterizing the intensity of interaction; nevertheless this factor can always be made equal to the value of 1 along with β in Equation (6) by selecting the appropriate units), whereas the second term describes the impact of the external magnetic field (*I*/*r_i_* and *φ_i_* represent its magnitude and direction respectively) which is produced by the current *I*. Then the solution of the motion Equation (6) (the initial state is set to *θ_i_* = 0: the magnetization distribution is initially homogeneous) corresponds to the magnetization reversal process, which is accompanied by an emission of a quasiparticle generated by four spins as shown in Figure 2 (only a part of the lattice is shown for the sake of clarity). After this quasiparticle leaves the model’s borders, all the spins end up oriented along the concentric circles, and the reverse magnetization process is completed.

It should be noted that the modeling approach proposed here is applicable only if the magnetization vector is guaranteed not to leave the plane of the sample, i.e., |*K*| → ∞. For any finite values of the anisotropy constant *K*, the orientation of the magnetization vector should be characterized by two angles, and in Formula (11), it is necessary to explicitly take into account the effect of the easy-plane anisotropy. In addition, the equation of motion (6) must be replaced by a more general equation containing the second derivative of the angles with respect to time. Such an extended model would make it possible, in particular, to study processes in which planar distributions of magnetization become unstable, as a result of which magnetization reversal processes proceed according to a qualitatively different scenario. However, the occurrence of such scenarios is obviously possible only for fairly small values of constant *K*. At the same time, calculations based on the use of Formulas (6) and (11) are not associated with significant computational resources. Thus, numerical experiments in this limiting case become extremely efficient, and their results can be considered a good zero approximation, which makes it possible to describe the mechanisms underlying the processes of the magnetization reversal of films with strong enough easy-plane anisotropy.

## 4. Magnetization Reversal Mechanism in the Case of Two Holes

As shown before, switching on a current in the filament passing through the hole leads to the nucleation of a quasiparticle with the topological charge +1, whereas the quasiparticle with the −1 charge is emitted in the direction which depends on the initial direction of magnetization in the sample. This scenario can be successfully used as a tool to analyze more complex situations, for example, when there is more than one hole in the sample. For instance, let the number of holes be equal to two, and magnetization in the homogeneous state of the sample is directed at the angle of π/4 to the line, which connects their centers. If the current is applied in one direction through both of the holes, two quasiparticles with −1 charges will be emitted. Let us point out, however, that neither of the particles can further move to infinity: in this case, the total topological charge of the sample would become +2, though, at a large distance from the holes, the two currents should be indistinguishable from one other and, therefore, the total charge (as in the case of a single conductor) imperatively equals +1. It follows that one of the emitted quasiparticles will actually never leave the sample area. This conclusion is effectively confirmed by a numerical experiment. Considering the discrete model comprising 392 spins with the energy similar to (11), we observe that the quasiparticle emitted by one of the holes changes the direction of its motion in practically no time and comes to a stop exactly between the holes, thanks to which a static structure emerges with the total topological charge of +1 (see Figure 3).

Of greatest practical importance is the situation in which only one antidot of the two available is energized (and, notably, the initial direction of magnetization is perpendicular to the line connecting the antidots). In this case, the quasiparticle with the −1 charge is emitted by one hole in the direction of the other, where it can be captured, which results in a bound state with the total topological charge of 0, which does not disappear when the current is removed. Figure 4 shows this stationary state formed by a short impulse of the current passed through the left-side hole. It should be pointed out here that the actual result of the effect of the current depends on its intensity and the impulse duration. If these are small, the quasiparticle fails to move substantially far away from the left-side hole by the time the current is switched off and thereafter, under the influence of the exchange interaction, moves back to the hole since the magnet becomes homogeneous again. If, however, the particle manages to cover more than half of the distance between the holes before the current is turned off, the exchange interaction contributes to its further movement so that there is no need to maintain the current pulse after this point; the particle will reach its stationary state without any external force. In the case of an extremely strong current, the particle, having reached the right-hand hole, overshoots it, and the film is reversely magnetized as a single whole instead of forming a bound state. Nevertheless, if the quasiparticle has no time to leave the sample area, it tries to go back to the source of its origin after the current is turned off and is captured by the right-hand hole, all the same forming a bound state. For this reason, the actual range of allowable current intensity values is quite extensive.

The practical value of the state illustrated in Figure 4 is that such an isolated magnetic inhomogeneity happens to be well localized and long-lived [21] thanks to which a system comprising two antidots can be used as a memory cell recording, which, as shown above, can be accomplished using short impulses of current. Moreover, in addition to the impulse already discussed, the current can be passed through the right-hand antidot as well as a result of which the +1 topological charge gets bound with the right-hand antidot, while the –1 one with the left-hand antidot, and the resulting state will become symmetrical to the one shown in Figure 4. Both of these situations can be restored to the homogeneous state: as an example, in the case of Figure 4, a sufficiently strong current should be passed toward the spectator through the left perforation. The described cell, therefore, can be switched over among the three relatively stable states (two symmetrical inhomogeneous and one homogeneous) thus implementing a rewritable trit. It can be deduced from this that a perforated magnetic film is a physical basis for creating storage media with ternary notation of data. The advantages of this approach versus the conventional binary notation are quote obvious and can bring about a significant increase in data density as well as a higher speed of organized access to the data.

## 5. Collision of Quasiparticles

Let us consider a film with two holes as in the case shown in Figure 3. This time, however, we shall pass the currents through the antidots in the opposing directions. In this case, each of the antidots will emit a single quasiparticle, the velocities of which will be opposing. Specifically, with certain initial directions of magnetization in the sample, the particles will move toward each other, which should result in their collision and in a subsequent movement into infinity since under the described configuration of currents, the total topological charge of the sample should, in a sufficiently long time, become equal to +2. Numerical modeling shows that in reality, the quasiparticles change the motion direction, even in those cases when one would think that their paths should not cross. For investigating this phenomenon, we shall revisit Equation (8) assuming *A* → 0 and $\Phi_{e}=-\frac{IM_{s}b}{2\pi r^{2}}\cos\left(\theta -2\varphi \right)$, which corresponds to the interaction of a magnet with the two opposing currents of the strength *I*, located at a small distance *b* one from the other (*r* >> *b*). Then, assuming *θ* = *θ*_0_ as the initial condition, similar to (10), we obtain
(12)$$\theta =2\varphi -2\arctan\left[\tan\left(\varphi -\frac{\theta_{0}}{2}\right)\exp\left(-\frac{\gamma Ibt}{2\pi \alpha r^{2}}\right)\right]$$

This ratio points to the existence of discontinuities of the function of *θ* on the half-lines $\varphi =\frac{\theta_{0}+\pi }{2}$ and $\varphi =\frac{\theta_{0}+3\pi }{2}$, which actually set out the direction of dispersion of the two quasiparticles. Thus, for instance, with *θ*_0_ = 0, dispersion of the particles after the collision should take place in the directions perpendicular to the line connecting the centers of the antidots, which is proven by calculations using a discrete model (see Figure 5).

## 6. Magnetization Reversal Mechanism in the Case of Four Holes

Similar to the case when vortex-like structures localize on two antidots, for these structures characterized by the +1 topological charge on one antidot and the −1 one on the other, magnetic inhomogeneities can localize on the four antidots located in the corners of a square with the ±1 charge associated with each of the antidots (see Figure 6). As shown in [21], such states possess additional bond energy as compared to the states localized on two antidots and display a greater stability in regard to thermal fluctuations. They therefore may turn out to be more suitable for developing ternary memory cell in practice. Nevertheless, the peculiarities of the magnetization reversal mechanism in the case of a four-antidot cell impose certain limitations on the cell geometry which have to be considered.

It is clear from the consideration of symmetry that for the origination of a vortex-like object shown in Figure 6, the opposing currents need to be passed through the two holes located on one diagonal (in this case through the upper left one and the lower right one). The selection of the diagonal here determines which of the two symmetrical inhomogeneous states will be eventually formed. As shown above, the magnetization reversal process in this case is connected with the dispersion of the two quasiparticles from the center of the system, and the angle of dispersion (shown with red arrows) forms the angle π/8 with the diagonal, connecting the centers of the two remaining holes (the blue dashed line in the figure). It follows from the fact that in this case, the direction *φ* = 0 happens to be rotated by the angle π/4 relative to the horizontal in the figure. This means that the initial direction of magnetization “upward vertically”, which is preserved at the border of the sample, is actually characterized by the angle *θ*_0_ = –π/4. For the bound state to be formed after the current is removed, the upper right and the lower left holes must capture the emitted quasiparticles, which means that the paths of the particles should at least cross these holes. Consequently, the angle at which each of the holes can be visible from the center should at least be π/4, which is attained with the radii of the holes $R\geq a\frac{\sqrt{2}}{2}\sin\frac{\pi }{8}\approx 0.27a$, where *a* is the distance between the centers of the neighboring holes. Thus, the space between the neighboring holes should not exceed 1.7*R*; otherwise, the emitted quasiparticles will pass the holes in an unrestricted manner, and the magnet always returns to the homogeneous state when the current is turned off. This conclusion is in good qualitative agreement with the results of the numerical modeling and means that though the bound states can be observed on the four holes of indefinitely small dimensions, their formation using the current impulses can be accomplished only when the dimensions of the holes and the cell itself are commensurate.

## 7. Sufficient Conditions for Implementing the Processes of Rerecording in Rewritable Memory Cells

As was earlier mentioned, for the inhomogeneous state to be formed in the cell with two holes, an impulse of current with a certain combination of intensity and pulse duration is necessary. The same applies to the cell with four holes if its geometry meets the determined requirements. Figure 7 shows the dependence of the minimal current strength *I*_m_ which is required for recording the inhomogeneous state into the cells of both types on the impulse duration τ (in conventional units). The calculations were made on the basis of a discrete model comprising 188 spins for the case of two holes and 308 spins for the case of four holes.

It can clearly be seen that the nature of the dependence presented in Figure 7 turns out to be similar for both types of the cells. Specifically, both graphs approach horizontal asymptotes. Their existence is connected with the fact that at the current intensities corresponding to the asymptotes, the systems turn into stationary states in which the emitted quasiparticles come to a stop at certain positions between the holes. Correspondingly, when an attempt is made to apply a somewhat smaller current, the particles will not reach these positions, whatever the time may be, and will return to the point of their origination after the current is turned off. If a somewhat stronger current is applied, the particles will reach the specified positions within a finite time, after which the current can be turned off, and capturing the quasiparticle by the proper hole will be accomplished due to the effect of the exchange interaction. Hence, the required current intensities are essentially limited at the lower side, though no such restrictions seem to be there as regards pulse duration.

## 8. Conclusions

It therefore follows from the above analysis that flat vortex-like inhomogeneities localized on the antidots of magnetic films can easily be controlled by impulses of current applied to some of these antidots. Since changes of the topology of an equilibrium structure in the absence of external influences become impossible, the inhomogeneities thus formed become long-lived and can be successfully used for data storage. Moreover, each cell of such magnetic memory codes not a bit but at least a trit of information since the state of a magnet in the vicinity of two or four antidots appears either homogeneous or one of the two inhomogeneous ones differing in the signs of topological charges. This opens up prospects for a significant boost of data density through a switch from the binary-coded representation of stored data to a much more efficient ternary representation.

It should be pointed out that the mechanism revealed in the paper of changing the topology of magnetic structures involving a transfer of the topological charge by quasiparticles is of interest on its own. Similar mechanisms can well be used for the development, based on the magnetic films not only of the data storage, but also the logical elements of nanoelectronics, including the creation of computer units completely implemented on magnetic principles without using semiconductors. In view of this, it is deemed important to further develop the ideas presented in the paper aimed at a full-fledged theory of motion and interaction of quasiparticles of the types considered here.

## Figures and Tables

**Figure 1 entropy-24-01104-f001:**
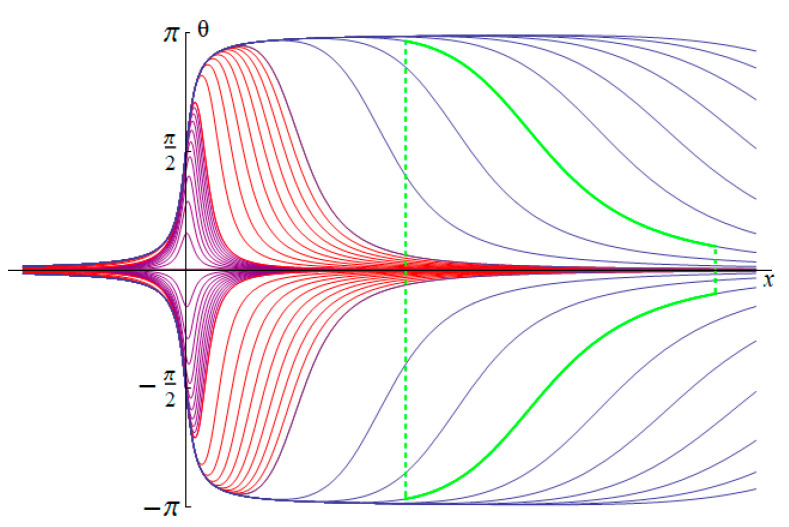
The curves showing the dependence of *θ* on *x* illustrating the process of magnetization reversal of the sample in the vicinity of the discontinuity line. The time lapse between subsequent blue curves is 10 times greater than that between subsequent red curves, which in its turn is 10 times greater than the time lapse between subsequent violet curves. The green solid and dashed lines outline the inhomogeneity area which can be regarded as an individual quasiparticle.

**Figure 2 entropy-24-01104-f002:**
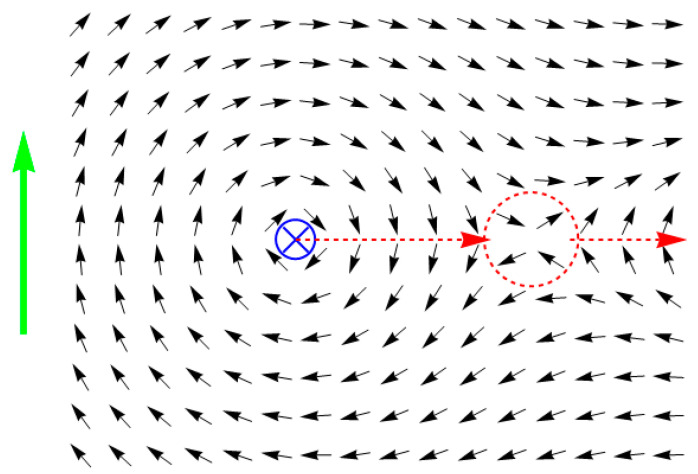
The magnetization reversal process which is accompanied by an emission of a quasiparticle. Here, the dashed lines in the form of a circle denote the area of the quasiparticle localization, whereas the dashed lines terminating in an arrow denote the path and direction of the quasiparticle motion. The initial state of the magnet is homogeneous; the green arrow shows the initial direction of the magnetization vector.

**Figure 3 entropy-24-01104-f003:**
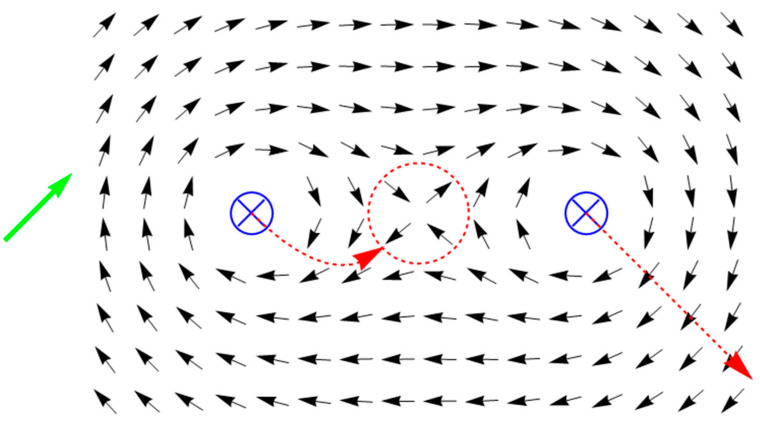
The magnetization reversal process with a stop of one of the quasiparticles. The meaning of the dashed lines and green arrow here is the same as in Figure 2.

**Figure 4 entropy-24-01104-f004:**
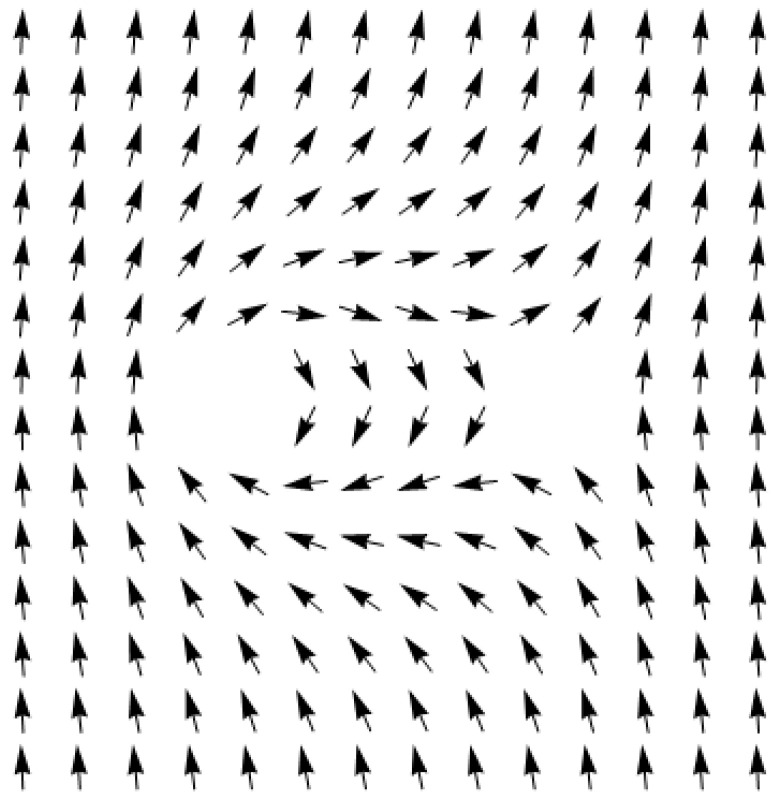
The bound state on two antidots.

**Figure 5 entropy-24-01104-f005:**
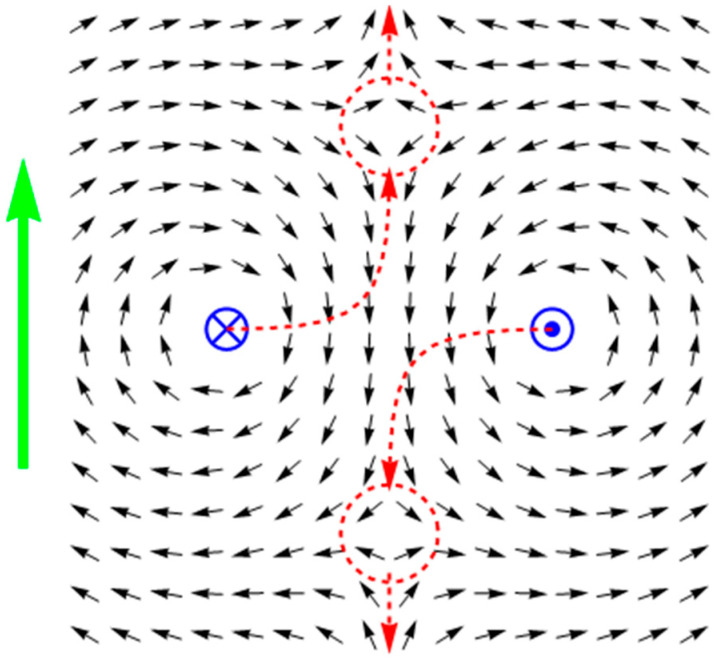
Dispersion of the quasiparticles after the collision. The meaning of the dashed lines and green arrow here is the same as in Figure 2.

**Figure 6 entropy-24-01104-f006:**
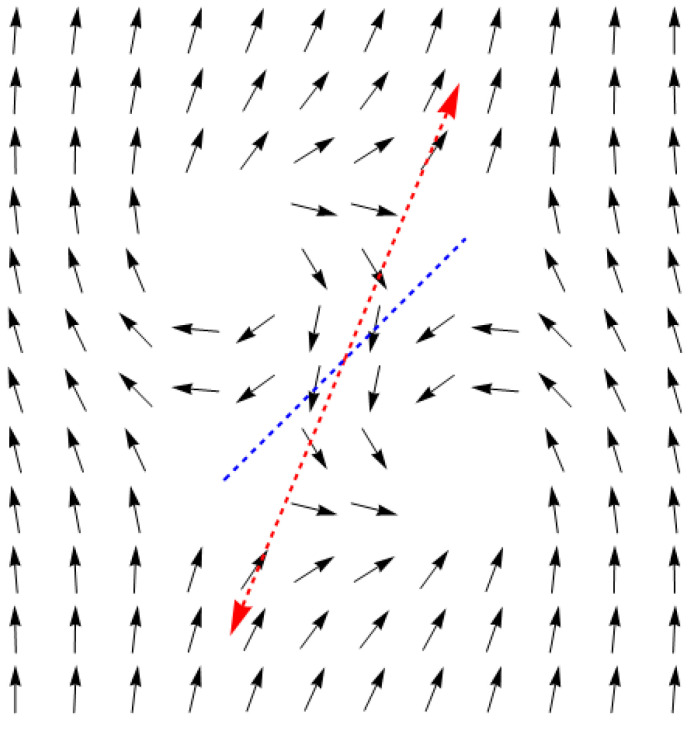
The bound state on four antidots and the mechanism of its formation. The blue dashed line here shows the diagonal connecting the centers of the two antidots, while the red dashed line terminated with arrows illustrates the direction of dispersion of quasiparticles.

**Figure 7 entropy-24-01104-f007:**
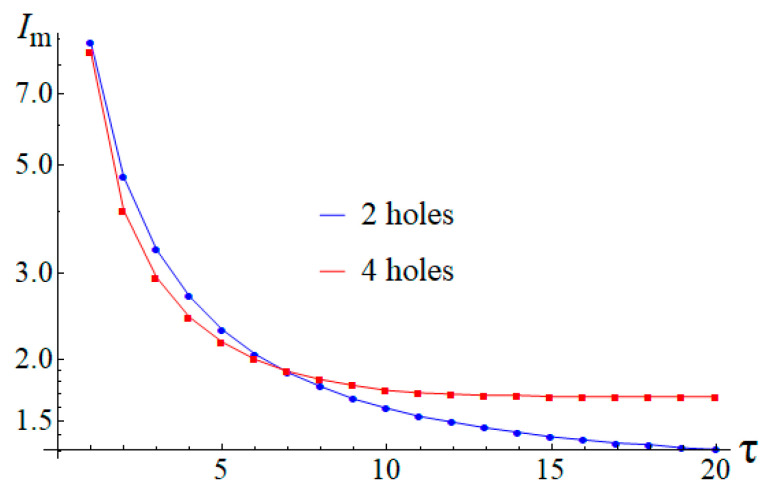
Dependence of the minimal current strength *I*_m_ on impulse duration τ. The blue line corresponds to the case of the two-hole cell, while the red line corresponds to the four-hole one.

## Data Availability

Not applicable.

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
