# Peer review of "Mechanism of Topology Change of Flat Magnetic Structures"

_entropy, 2022, doi:10.3390/e24081104_

Round 1
Reviewer 1 Report
The paper deals with the dynamics of multi-vortex textures created in planar (in-the-plane-ordered) magnets around normal-to-the-plane current-carying filaments.
The formulation of the dynamical problem and the description of the results are quite unclear. The initial states for the simulations of Sections 3-6 should be precisely defined. To improve the clarity, I would suggest to provide the snapshots of the dynamical processes in series, in a manner of e.g. A. Janutka, IEEE Magn. Lett. 4 (2013) 4000304, incuding at least the intitial and perhaps final (asymptotic) states.
The formulation of the model is shortened relative to Ref. [6], unfortunately, becoming unclear. In particular, the definition of the angle 'varphi' (characterizing the orientation of the Oersted field) should be provided. Moreover, it is unclear what is the meaning of 'varphi' in the case of the presence of several current-carying fibers (Sec. 5), (in Ref. [6]. the authors have introduced a series of 'varphi_i' angles of the index 'i' enumerating the fibers).
A vast available literature on the magnetic-vortex structures is omitted, while, the authors address the literature on the magnetic skyrmions which is not especially relevant to the systems considered. Indeed, skyrmions are widely believed to be applicable to storing the information with ultimate density. To the best of my knowledge, the vortices aren't currently. Hence, the motivation for the study requires to be strenghtened.
Author Response
We did our best to take into account carefully all the comments of the reviewers. As a result, the article has been significantly revised. Here is a list of improvements in accordance with the comments of the reviewers:
Reviewer 1.
1.1. The formulation of the dynamical problem and the description of the results are quite unclear. The initial states for the simulations of Sections 3-6 should be precisely defined. To improve the clarity, I would suggest to provide the snapshots of the dynamical processes in series, in a manner of e.g. A. Janutka, IEEE Magn. Lett. 4 (2013) 4000304, incuding at least the intitial and perhaps final (asymptotic) states.
It was indeed rather unclear from the text that all the initial states were considered to be homogeneous. To clarify the situation, we included an explicit remark into the text and also modified figures 2, 3, 5. Now there is a green arrow on the figures showing the initial direction of the magnetization vector (a corresponding comment was added to the figure labels). However, adding a separate figure for the initial state which is homogeneous looks to be redundant, so we decided not to include them after all (nevertheless it’s only the proposal and we obviously could add these figures if the reviewer still finds them to be of use).
1.2. The formulation of the model is shortened relative to Ref. [6], unfortunately, becoming unclear. In particular, the definition of the angle 'varphi' (characterizing the orientation of the Oersted field) should be provided. Moreover, it is unclear what is the meaning of 'varphi' in the case of the presence of several current-carying fibers (Sec. 5), (in Ref. [6]. the authors have introduced a series of 'varphi_i' angles of the index 'i' enumerating the fibers).
We added an extensive explanation of formula (5) (it’s equation (11) now) as well as a clarifying remark for the energy terms leading to (4) (it’s (10) now). We also included a “reminder” for the varphi meaning in the case of the continuous model and changed varphi to varphi_i in the case of the discrete model, exactly as the reviewer suggested. It really looks much better now.
1.3. A vast available literature on the magnetic-vortex structures is omitted, while, the authors address the literature on the magnetic skyrmions which is not especially relevant to the systems considered. Indeed, skyrmions are widely believed to be applicable to storing the information with ultimate density. To the best of my knowledge, the vortices aren't currently. Hence, the motivation for the study requires to be strenghtened.
The list of references has been vastly extended (from 8 to 24). So has been the introduction: about 1 extra page added to clarify the motivation of the study and its place among the existing studies.

Reviewer 2 Report
The work “Mechanism of topology change of flat magnetic structures” by E. Magadeev et al. is not immediately publishable. First, although the journal admits a general profile, I would say that, given the specialized aspects in magnetism, the “Entropy” is not the perfect place.
However, in order to have it published in this journal, the work must be rearranged.
The introduction should be extended, including concrete examples of materials exhibiting the discussed properties and phenomena.
The actual reference list, with 8 entries is far insufficient for a decent work. There are many other works dealing with magnetization reversal to be considered and correlated with actual results.
The premises of the modelling are quite cryptic. The equation (5) from manuscript seems pivotal to this work, but it is abruptly posed. According to the nearby text, it accounts coupling, but then, which is the associated parameter. From where the 1/r phenomenology comes?
For this equation, the authors are offering reference [6] which is a recent reference of their own, in JETP letter. I cannot reach this work and I am afraid that a certain overlap with the actual material may exist.
From another work of the authors, written in Russian (which I cannot follow), posted in arXiv the above mentioned equation is cited with a Russian textbook from 1973. Then, the phenomenological premises should be explained and demonstrated more clearly, in a consistent and complete manner.
The limits and drawbacks of the modelling should be clearly admitted and presented.
The manuscript is written quite negligently: the abstract contains improper terms antidote/antidotes, while the main text has the correct couple antidot/antidots (of course being not the case of medical term “antidote”). Many equations are not numbered, some of them are unnecessarily written in bold characters.
Although, hopefully, the core of the work may be kept with the same message and case studies, a major revision should reshape the article.
Author Response
Reviewer 2.
2.1. The introduction should be extended, including concrete examples of materials exhibiting the discussed properties and phenomena.
The introduction has been vastly extended. The concrete examples of materials have been added in the beginning of Section 1.
2.2. The actual reference list, with 8 entries is far insufficient for a decent work. There are many other works dealing with magnetization reversal to be considered and correlated with actual results.
The list of references has been vastly extended (from 8 to 24). The review on other works has been extended by about 1 extra page.
2.3. The premises of the modelling are quite cryptic. The equation (5) from manuscript seems pivotal to this work, but it is abruptly posed. According to the nearby text, it accounts coupling, but then, which is the associated parameter. From where the 1/r phenomenology comes?
A thorough explanation has been added near the equation (5) (it’s equation (11) now).
2.4. For this equation, the authors are offering reference [6] which is a recent reference of their own, in JETP letter. I cannot reach this work and I am afraid that a certain overlap with the actual material may exist. From another work of the authors, written in Russian (which I cannot follow), posted in arXiv the above mentioned equation is cited with a Russian textbook from 1973. Then, the phenomenological premises should be explained and demonstrated more clearly, in a consistent and complete manner.
We are very happy to provide the reviewer with the English version of the article in JETP Letters (attached to the e-mail). It has absolutely no overlap with the actual material: that article was completely devoted to the statical properties of the inhomogeneities, yet the paper under consideration deals with the dynamics. The material is brand new and has been never published before, even partially.
2.5. The limits and drawbacks of the modelling should be clearly admitted and presented.
A large paragraph has been added to the ending of the Section 2. It discusses cons and pros of the presented approach.
2.6. The manuscript is written quite negligently: the abstract contains improper terms antidote/antidotes, while the main text has the correct couple antidot/antidots (of course being not the case of medical term “antidote”). Many equations are not numbered, some of them are unnecessarily written in bold characters.
The abstract has been corrected, as well as several nasty typos here and there. All the equations have been numbered. The formatting has been corrected.

Round 2
Reviewer 1 Report
The authors have taken into account my suggestions. I believe the present version to be improved relative to the previous one. Some minor comments are the following.
Eq. (11) is written with relevance to a single antidot placed at the center of the refernce frame. Since the authors refer to (11) considering many-antidot cases, perhaps, introducing summation over the antidots to the second (Oersted-field) term would make it clearer (thus, r_i -> r_i-R_n, varhi_i -> varphi_i-phi_n, I -> I_n, where n would index the antidot). In case, the authors would decide not to change (11), I would suggest to write it explicitely that (11) corresponds to the presence of just one antidot in the center of the rerence frame.
In Fig. 3, it is unclear what is shown with the right-hand-side dashed line (with arrow) which begins at the right-hand-side antidot. Is the right vortex moving in the plane?
I've noticed some misprints:
- page 4, between (8) and (9): "plays but a stabilizing role".
- page 5, below (30): "is 10 greater than that between".
Reviewer 2 Report
The authors completed the manuscript in the critical aspects, reaching the limit of being publishable.